# Urban form and its impacts on air pollution and access to green space: A global analysis of 462 cities

**Nazanin Rezaei**[1]*, **Adam Millard-Ball**[2]

**1** Department of Environmental Studies, University of California, Santa Cruz, Santa Cruz, California, United States of America, **2** Department of Urban Planning, University of California, Los Angeles, Los Angeles, California, United States of America

* nrezaei@ucsc.edu

**Data Availability Statement:** The data underlying the results presented in the study are available at Dryad: https://doi.org/10.7291/D13H47 Here is the link to the code uploaded to GitHub: https://github.com/nazanin87/Urban-Form-Metrics.

## Abstract

A better understanding of urban form metrics and their environmental outcomes can help urban policymakers determine which policies will lead to more sustainable growth. In this study, we have examined five urban form metrics–weighted density, density gradient slope, density gradient intercept, compactness, and street connectivity–for 462 metropolitan areas worldwide. We compared urban form metrics and examined their correlations with each other across geographic regions and socioeconomic characteristics such as income. Using the K-Means clustering algorithm, we then developed a typology of urban forms worldwide. Furthermore, we assessed the associations between urban form metrics and two important environmental outcomes: green space access and air pollution. Our results demonstrate that while higher density is often emphasized as the way to reduce driving and thus $PM_{2.5}$ emissions, it comes with a downside–less green space access and more exposure to $PM_{2.5}$. Moreover, street connectivity has a stronger association with reduced $PM_{2.5}$ emissions from the transportation sector. We further show that it is not appropriate to generalize urban form characteristics and impacts from one income group or geographical region to another, since the correlations between urban form metrics are context specific. Our conclusions indicate that density is not the only proxy for different aspects of urban form and multiple indicators such as street connectivity are needed. Our findings provide the foundation for future work to understand urban processes and identify effective policy responses.

## 1. Introduction

Urban form is one of the most central topics in the study of urbanism. Once the form of a city has been defined, it is challenging and expensive to alter it significantly [1]. The physical shape and structure of cities may lead to adverse environmental impacts such as greenhouse gas emissions, economic effects such as high public service costs and related taxes, and social and political consequences like inequality and social segregation. Therefore, urban form has become a topic of widespread public policy and academic interest [2–4]. Moreover, progress towards smarter and more sustainable growth can only be made with a complete

**Funding:** The author(s) received no specific funding for this work.

**Competing interests:** The authors have declared that no competing interests exist.

understanding of urban form and its relationship with air pollution [5, 6] and other environmental outcomes, as well as the policy measures required to respond to it.

Recent trends such as the availability of high-quality spatial data, access to aerial photographs and satellite images, the use of spatial analysis software packages, and geographic information system (GIS) technology have significantly contributed to the interest in studying urban form and progress in developing its measures [7, 8]. A variety of spatial metrics have been developed to characterize and quantify urban form [9–15]. The selection of the proper metrics requires an understanding of the available data, the scope and discipline of the research, and the characteristics of the geographical region.

Initially, density was the primary metric used to measure and characterize urban form, and it still is since it can be measured relatively easily and data is readily available. Many studies point to its association with vehicle ownership and travel [13, 16]. In addition, density is assumed to correlate with other aspects of urban form such as size, coverage, and compactness and is easy to understand and interpret. However, it is not evident whether density is a reasonable proxy for these different aspects of urban form in various contexts, or whether multiple indicators are needed. There are also problems with simple density measures since they may not accurately reflect the actual density faced by the individuals or firms involved. As an example, the metropolitan area of Flagstaff, Arizona, consists of the second-largest county in the U. S., but is spread across multiple forests, national parks, and monuments, meaning that the density experienced by the average person is much higher than average density as typically measured (people per square kilometer) [17].

Where data are available, researchers have therefore gone beyond density and considered alternative and/or complementary measures. For example, Burchfield et al. [11] use the percentage of undeveloped land in a square kilometer surrounding an average residential area as an indicator of urban form. Another group of researchers has used principal component analysis to combine many variables into higher-dimensional factors such as degree of mixed-use, degree of centrality, and street accessibility to operationalize urban form [18]. Urban form can also be measured by per capita land consumption, i.e., land area divided by population, which demonstrates the efficiency of growth in urban areas [19]. Recently, Barrington-Leigh and Millard-Ball [15] have introduced a measure of urban form called the street-network disconnectedness index (SNDi). Their study demonstrates that connected street networks, such as grids, increase accessibility by walking, bicycling, and public transportation, as more destinations can be reached in a specific time. In contrast, dendritic or cul-de-sac-dominated networks favor traveling by private cars [15]. Furthermore, Sevtsuk and Amindarbari [12] defined nine metrics of urban form in their study: size, density, coverage, discontiguity, compactness, polycentricity, expandability, and land-use mix.

Patterns of urban form are extensively studied in the West, particularly the U.S. [20]. In recent years, the shape of urban growth has also become a worldwide concern; however, it is unclear whether the results from studies in the U.S. generalize globally. It is useful to focus more on this area of research because cities are expanding their boundaries faster than their populations, and unplanned development is mainly related to the cities in developing countries that have been unprepared for absorbing the high number of often-impoverished rural people [14]. Research outside the U.S. has begun to emerge, often relying on North American sprawl concepts to describe trends in Europe, China, and even diverse locations such as India [21], Iran [22–25], Japan [26], Turkey [27, 28], and Egypt [23]. The results of one recent review paper in this field show a clear unevenness of the geographic coverage since 85% of their reviewed articles focus on Europe, North America, or China [29]. Moreover, there is limited comprehensive, comparative, and global-scale research on cities despite their significant role in economic, political, and environmental systems around the world [30].

Among the previous studies in this area with global coverage are Atlas of Urban Expansion [14], Global Comparative Analysis of Urban Form [31], and the Global Homogenization of Urban Form [32]. Huang et al. [31] have used 77 satellite images of metropolitan areas measuring spatial metrics such as complexity, centrality, compactness, and density. Atlas of Urban Expansion uses a sample of 200 cities from the UN Global Sample of Cities to measure some important attributes of cities such as density, compactness, and fragmentation. Similarly, Lemoine-Rodriguez et al. [32] have applied landscape metrics such as ratio of open space, patch density, edge density, and landscape shape index to a global sample of 194 cities obtained from the Atlas of Urban Expansion database.

This paper builds on this literature by using a global dataset to first develop a typology of urban forms worldwide. As it is unclear how various measures of urban form relate to each other in different parts of the world, we compared different urban form metrics and examined how they co-vary with each other and across geographic regions and socioeconomic characteristics such as income. Second, we examine the relationship between our typology and two important policy outcomes: access to green space and air pollution, specifically $PM_{2.5}$.

Fine particulate matter ($PM_{2.5}$) refers to suspended particulates smaller than 2.5 microns in diameter capable of penetrating very deep into the respiratory tract [33]. Among all air pollutants, $PM_{2.5}$ poses the greatest health risk globally, affecting more people than any other pollutant and increasing the risk of respiratory and cardiovascular diseases in chronic exposure. $PM_{2.5}$ can be of natural (i.e., sand and dust) or of anthropogenic source (i.e., combustion residuals), and its concentration is of high concern especially in urban agglomerations that are dense and growing rapidly [34]. Two measures are important to consider: population exposure to air pollution by fine particulates and emissions of fine particulates from transportation. While the two measures are related, the first emphasizes the health *impacts*, while the second emphasizes the *contributions* of the transportation sector, given that urban form is most likely to affect emissions through its impacts on vehicle travel.

Access to green spaces in urban areas has been recognized as an essential component of the urban environment [35]. Green spaces in cities mostly include semi-natural vegetation cover, such as street trees, lawns, parks, gardens, forests, green roofs [34]. Therefore, this study aims to find the relationship between urban form and access to green spaces in large urban areas using two measures: green space per capita and the share of population living in areas of high green.

The following section reviews the literature on measuring urban form, mainly focusing on weighted density, density gradient, compactness, and street connectivity metrics. We then discuss our methods, including data characteristics, computational methods, and analysis methods such as Lowess regression, K-Means clustering, and random forest regression. Afterward, we present the results and discussion section examining correlations between urban form metrics, typology of the cities, the geographic distribution of the clusters, and environmental outcomes of urban form metrics (Fig 1). Finally, we conclude by suggesting that while higher density is often emphasized as the way to reduce driving [36, 37] (due to higher access to active and public transit) [38, 39] and thus $PM_{2.5}$ transportation emissions, it comes with a downside: less green space access, and more $PM_{2.5}$ exposure. Moreover, street connectivity has a stronger association with reduced $PM_{2.5}$ emissions.

## 2. Measuring urban form

Urban form refers to the physical characteristics of built-up areas, such as the shape, size, density, and configuration of settlements [40, 41]. Due to its negative environmental, social, and economic impacts, urban sprawl is considered as the unfavorable display of urban form. While there are many definitions and dimensions of sprawl, sprawl is used to describe a process with the result of

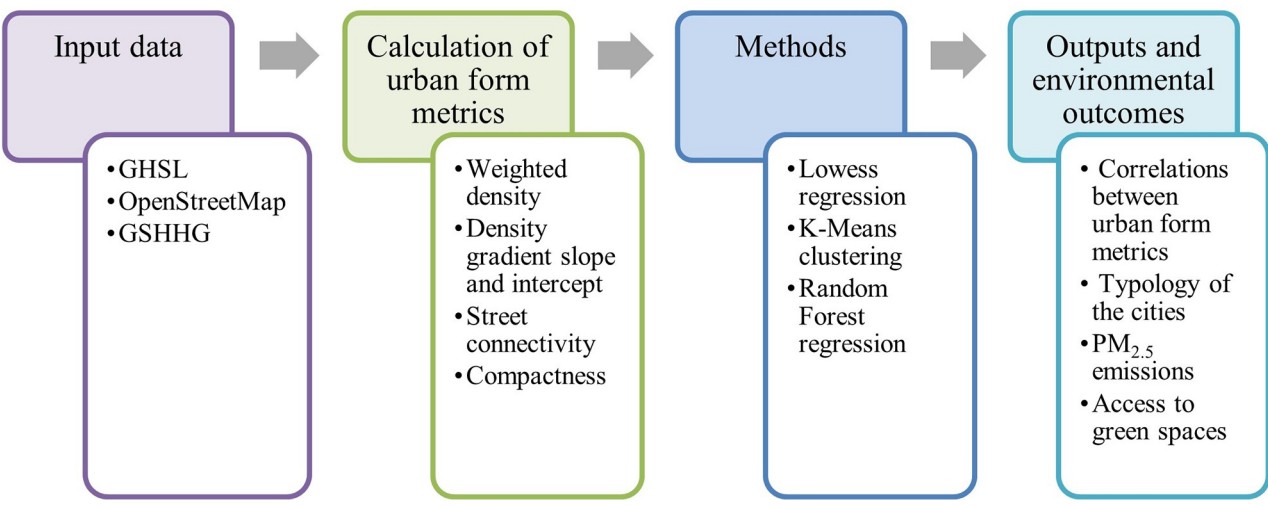

**Fig 1. Flowchart of the input, process, and output of the study.**

low-density, scattered, leapfrog or isolated development, unplanned and haphazard land-use, end-less commercial strip malls, and the lack of adequate transportation and housing options with high levels of automobile reliance at the periphery of urban areas [3, 14, 18, 42–44]. It describes cities where people need to drive long distances to conduct their daily lives or where employment is decentralized [45]. According to Galster et al. [9], sprawl is a pattern of land-use in an urban area that exhibits insufficiencies within some or all of eight distinct dimensions, including continuity, density, centrality, concentration, clustering, nuclearity, mixed uses, and proximity. In this paper, we prefer the term "urban form" as it is more general and less value-laden than "urban sprawl", but our analysis could be interpreted as a study of urban sprawl.

Urban geographers, planners, economists, and other social scientists have developed numerous measures to quantify urban form characteristics, typically using remote sensing and/or census-based population datasets. In this section, we review the literature and discuss the motivation for several of these measures, focusing on the ones we employ in our analysis and treating others more briefly. The metrics of urban form that are measured in this research include street connectivity, weighted density, density gradient (intercept and slope), and compactness. Street connectivity is selected due to the major long-lasting impact of street networks on urban form. Density is the metric used in most similar global studies, and we use a variant, weighted density, in this research. Density gradient is another descriptive and analytical metric that demonstrates the change in population density across the city. Compactness is selected as it reflects the ratio of built-up area to the buildable area within the convex hull and is an important metric used in urban form studies. A brief definition of each measure is explained in Table 1.

**Table 1. Urban form metrics used in this research [12, 15, 17, 46].**

| Name of the Metric | Definition |
| --- | --- |
| Street Connectivity | We use the Street Network Disconnectedness Index (SNDi), which is a summary scalar measure for street-network sprawl |
| Weighted Density | The density in each one kilometer square grid cell, weighted by the population of that grid cell |
| Density Gradient | The change in population density in an urban area from the center to the periphery |
| Compactness | The built-up area of the city divided by the area of its convex hull (excluding unbuildable areas) |

## 2.1. Street connectivity

Previous research in Europe and North America shows that disconnected urban street networks are strongly associated with increased vehicle travel, energy use, and $CO_2$ emissions. In one of their recent articles, Barrington-Leigh and Millard-Ball [47] introduced a summary measure of street-network sprawl called the street-network disconnectedness index (SNDi). They used a large dataset of intersections and edges to calculate nodal degree for each intersection, dendricity, circuity, and sinuosity. Dendricity shows whether "a network is tree-like in that there is only one possible route to reach certain nodes". Circuity means "the ratio of path length to straight-line distance between each node and other nodes in its vicinity". Sinuosity is the ratio of length to end-to-end distance for every edge. They used a principal components analysis to assign empirical weights to their aggregate measures. They focus on the first principal component as an overall measure of street-network sprawl and refer to this measure as the Street-Network Disconnectedness Index (SNDi) [47].

## 2.2. Weighted density

Density is typically calculated as the number of people in a given geography (city, country, grid cell, etc.) divided by the land area. However, such a "simple density" measure may not accurately reflect the actual density faced by the individuals or firms involved. For example, the density of Canada does not reflect the experience of someone in central Toronto or Montreal. Moreover, density depends on the arbitrary shape and size of the units of analysis, such as whether parks, deserts, or other open spaces are included. Duranton and Puga [17] propose that we address these challenges by measuring "experienced density" instead, i.e., by counting the population within a given radius around each individual. In addition to addressing the arbitrary nature of boundary lines, such experienced density also captures how close the typical individual is to others in an unevenly distributed population. In practice, experienced density is not feasible to calculate as the precise location of each individual is not known, but a weighted density measure is similar in practice–for example, weighting grid-cell level population densities by the population of that grid cell. It is essential to weight by population since otherwise, we would be calculating population within ten kilometers of the average place instead of within ten kilometers of the average person [17].

## 2.3. Density gradient

Population density gradient has been extensively used in urban studies, both as a descriptive device and as a hypothesis-testing instrument. This measure represents a useful summary statistic of the distribution of population concentration in cities [48, 49]. In other words, it indicates the unevenness of population distribution. As density gradients incorporate crucial aspects of urban land-use and overcome traditional constraints in measuring urban densities, they are potentially helpful for measuring urban sprawl. Moreover, the calculation of density gradients is generally independent of the dynamics of urban structures, such as shifts in the urban periphery and changes in its boundary over time, since they use the distance from activity centers as their basis [50].

Population density gradient depends on the spatial distribution function. According to Parr, the negative exponential function is a better fit for describing density in urban areas, while the inverse power function is better suited to the urban fringe and hinterland [51]. Urban scholars often associate population density gradients from a negative exponential function with Alonso, Muth, and Mills' pioneering work, but it was actually Colin Clark who popularized this concept first in 1951 [52]. Angel et al. [46] calculated density gradient by measuring the population density of small administrative areas. As a next step, they found the

average population density in rings of increasing distance from the city center. Finally, they fitted a negative exponential curve to the average ring density points as a function of distance from the city center. The gradient is steep when a city's population density falls rapidly as the distance from the city center increases. When population density is pretty much the same in the center and everywhere else, the gradient is shallow [46, 50].

## 2.4. Compactness

Compactness is considered as one of the most important aspects of geographic shapes and can be quantified in several ways [53, 54]. A common approach to calculate compactness is to measure the ratio of area to a circle [55] or the polygon's convex hull [56]. According to Angel et al. [46], compactness is defined as the ratio of the observed built-up area to the observed buildable area within "the circle of minimum radius encompassing the consolidated built-up area of the city".

Sevtsuk and Amindarbari [12] have proposed two new size metrics to consider when calculating the size of the urban centers. One is the area of the convex hull around all developed polygons. The convex hull is defined as "the smallest flat polygon that has no concavity in its perimeter and that fully contains all individual polygons of the urban extent of a city". The second metric is the unbuildable area within the convex hull. The authors suggest that the definition of "unbuildable" should include areas that are not buildable due to natural barriers, such as water bodies, steep slopes, or other natural limitations, but exclude those that result from policy decisions (e.g., Central Park in Manhattan). This definition will allow for a more reliable and consistent assessment of density [12].

Built-up coverage shows how much of the total urban extent or a particular sub-area of the city is occupied by built-up areas. Estimating coverage within the convex hull minus unbuildable area is a consistent way to measure how much land is urbanized within a city. The exact shape of the hull and the extent of the unbuildable area varies from city to city. However, the estimation remains consistent and comparable since remaining vacant land within the hull can be said to be vacant voluntarily [12]. We consider the built-up coverage metric the same as compactness in our study. There are other definitions of compactness, but we use the convex hull method as it is flexible in cities with different shapes and topographical and water constraints. This metric is commonly estimated for building footprints—the percentage of the urban area covered by buildings.

Compactness is scale-dependent and image resolution can affect the absolute value of our compactness measure [57]. However, we expect scale dependency of this metric to have limited impact on our results since we use a consistent image resolution across the study, and we examine the difference in compactness between the cities rather than the absolute value.

## 2.5. Other metrics

A number of other metrics have been used and presented in various studies, such as polycentricity, land-use mix, built-up density, and fragmentation. We discuss some of them briefly here, but do not use them in the empirical portion of this paper due to a lack of appropriate data or similarity to the other metrics selected.

**2.5.1. Polycentricity.** Polycentricity describes the degree of employment concentration in a city's sub-centers. This metric can also show how many significant job centers are within those sub-centers, what their size distribution is, and what employment percentage is located there. Polycentricity is hard to consistently quantify since the definition of a center is not obvious and the number of centers cannot necessarily be the sole indicator of polycentricity [12]. Moreover, calculating polycentricity requires access to high-resolution data on employment,

which is not available at the global level. While we experimented with a calculation of residential polycentricity using population data, the results did not appear meaningful or relate to employment-based polycentricity measures.

**2.5.2. Land-use mix.** The planning literature has extensively discussed the effects of land-use mix on traffic congestion, transportation energy consumption, real estate values, and crime rates. Two popular types of metrics are used to capture land-use mixing: one is based on the number of different uses found in a given area, making it possible to rank areas according to the number of different land-uses they support, and the second evaluates a land-use pattern's heterogeneity or homogeneity by examining how equally each use occupies an area [12].

Hamidi et al. [57] proposed three variables for calculating land-use mix using principal component analysis: a measure of job-population balance, a measure of job mix, and an average weighted walk score for metropolitan areas. According to Sevtuk and Amindarbari [12], an optimal land-use mix metric function should be estimated at the level of small intra-urban subdivisions rather than the entire city. The most accurate estimation is possible when the land-use data is given at pixel or raster cells level. We did not include the land-use mix metric in our study since we did not have access to any global datasets of employment numbers.

**2.5.3. Urban extent density and built-up density.** Urban extent density is the ratio of the total population of the city to its urban extent, measured in persons per hectare. Built-up area density is the ratio of the total population of the city to its built-up area, measured in persons per hectare. Urban extent density is always lower than built-up area density. Also, because the urban extent of the city contains its urbanized open space, urban extent density is not independent of the city's level of fragmentation, while built-up area density is. Two cities with the same population and the same built-up area will have the same built-up area density. If one city is more fragmented—its built-up area occupies only 40% of its urban extent—and the other city's built-up area occupies 80% of its urban extent, then the urban extent density in the former will be half that of the latter [14].

**2.5.4. Fragmentation.** Fragmentation measures to what extent a city's built-up area covers its urban extent or, how much built-up area within an urban extent is fragmented by urbanized open space. A more fragmented built-up area implies a lower urban extent density, greater distances between places within the city, and more open space disturbed. Angel et al. [14] provide two measures of urban fragmentation that highly correlate with each other. One is the ratio of the built-up area within the urban extent of the city and its urban extent, called saturation. The other is openness, which is "the average share of open space pixels within the Walking Distance Circle (a circle with an area of 1km$^2$ and a radius of 564 meters) of every built-up pixel within the urban extent" [14]. We decided not to use metrics such as built-up density and fragmentation as our measure of compactness addresses a similar aspect of urban form.

**2.5.5. Different geometric measures.** There are various geometric metrics that can be used to measure the shape of cities, such as circularity ratio, form ratio, elongation ratio, ellipticity index, and radial shape index. Circularity ratio, proposed by Miller in 1953, is calculated by multiplying the urban area by four and dividing it by the square of the perimeter (the length of urban boundary). Form ratio, suggested by Horton in 1932, is the ratio of area to the square of the length of the longest axis of a region. Elongation ratio, proposed by Werrity in 1969, is defined as the ratio of the length of the longest axis of a region to the length of its secondary axis. This ratio measures the extended degree of a region; the more extended the urban shape, the higher the ratio [58–61].

The ellipticity index, proposed by Stoddart in 1965, is the inverse of the compactness ratio. It is measured by dividing the area of the smallest circle to enclose the figure by the urban area [59, 61]. The Boyce-Clark radial shape index measures shapes by comparing the relative lengths of regularly spaced radials extending outward from a central node to the values a circle

would have [62]. Since these measures are similar in spirit to compactness and it is unclear whether they would have additional explanatory power with respect to outcomes such as emissions and green space, we have not used them in our study. However, using a wider range of urban form measures could be a topic of future work.

## 3. Methods

### 3.1. Data

Our primary data are drawn from the European Commission Joint Research Center's Global Human Settlement Layer (GHSL), which contains fine-scale global and multitemporal geospatial data on populations and built-up areas worldwide. By using GHSL, cities can be studied globally in a consistent and comparative way. By looking at population dynamics and the expansion of built-up areas, GHSL analyzes the growth of human settlements worldwide [63].

GHSL data are available for 1975, 1990, 2000, and 2015 at three different scales: 30m, 250m, and 1km. The one-kilometer grid cells were used to calculate urban form metrics in this study. The GHSL project produces global population density grids (GHS-POP) by combining the built-up area grid (GHS-BUILT) with census data through spatial modeling techniques. The population data is collected by national censuses with heterogeneous criteria and heterogeneous update time [34]. One novelty of our study is using the GHSL data to calculate different measures of urban form and compare them across the urban centers worldwide.

The GHSL dataset defines 13135 Urban Centers (UCs) worldwide. A UC is roughly equivalent to a city, but the boundaries are based on population and built-up intensity rather than administrative boundaries. UCs are spatially delineated by a "degree of urbanization" model, which considers population size, density, and contiguity of populated grid cells. Each UC is a cluster of grid cells such that each $1km^2$ grid cell has a density of at least 1,500 inhabitants, or 50% of the surface is built-up, and the cluster of grid cells is contiguous with a minimum population of 50,000 inhabitants. In this study, urban centers with more than 1 million population in 2015 were used, which includes 462 cities.

About 56 percent of the cities above 1 million studied in this research are located in Asia. 13.5 percent in Africa, 12 percent in Latin America and the Caribbean, 10 percent in Europe, 7.5 percent in Northern America, and less than one percent in Oceania. Around 23 percent of the cities above 1 million population are in high income countries, 34 percent are in upper-middle-income countries, 38 percent are in lower-middle-income countries, and 6 percent are in low-income countries.

The GHSL dataset also provides our two outcome measures: $PM_{2.5}$ and access to green space. In the GHSL data, $PM_{2.5}$ emissions are derived from the European Commission's in-house Emissions Database for Global Atmospheric Research that estimates anthropogenic greenhouse gas and particulate air pollutant emissions from 1970 to 2012 [34]. The calculation of the emissions considers all human activities, except large-scale biomass burning and land use, land-use change, and forestry. Due to the bottom-up compilation methodology of sector-specific emissions applied consistently across all world countries, the results are comparable. Total emissions of $PM_{2.5}$ from the transport sector in 2012 is used in this study, which is calculated based on 2015 urban center boundaries, and expressed in tonnes per year. This measure is used to calculate the transportation emissions per capita. In addition, $PM_{2.5}$ concentration, based on the Global Burden of Disease (GBD) 2017 data on ambient air pollution from 1990 to 2015, is used to determine emissions exposure. This metric is expressed in $μg/m^3$. As the $CO_2$ emissions data is available only at the country level, and not at the city level, we did not include it in this study.

The data on green spaces has been produced by analyzing Landsat annual Top-of-Atmosphere (TOA) reflectance composites created by considering the highest value of the Normalized Difference Vegetation Index (NDVI) as the composite value (i.e., greenest pixel). Average greenness is estimated for 2014 and calculated based on the 2015 urban center boundaries. The access to green spaces is measured through a proxy metric, "generalized potential access to green areas", which measures the number of people living in high green areas at the generalization scale of the spatial data used in the assessment, regardless of their usability. The metric builds on the greenness metric derived from remote sensing Landsat imagery. The area of high green in 2015 is delineated per each urban center, and then intersected with the population grid of epoch 2015. Finally, the share of the urban center population living in this area is estimated from the total population of the urban center [34].

In addition to GHSL, we use three other data sources. First, the country classification by income level is based on 2016 GNI per capita from the World Bank. When it comes to income, the World Bank divides the world's economies into four income groups: high, upper-middle, lower-middle, and low [34]. Second, shoreline data are from the Global Self-consistent, Hierarchical, High-resolution Geography (GSHHG) database, developed by Wessel and Smith and updated in June 2017 (https://www.soest.hawaii.edu/pwessel/gshhg/). We use the shoreline dataset to calculate the unbuildable areas within a convex hull to measure compactness. Third, street connectivity data are from Barrington-Leigh and Millard-Ball [15], updated with the February 2022 vintage of the OpenStreetMap road network.

## 3.2. Computational approach

Our selected urban form metrics were calculated using SQL code from data stored in PostgreSQL/PostGIS database. Here, we provide a precise definition of each metric.

Street connectivity is measured by the Street Network Disconnectedness Index (SNDi), and we do not make further calculations here beyond aggregating to urban center boundaries. SNDi is an index that includes measures of circuity, the proportion of deadends, and other connectivity measures, as explained in Barrington-Leigh & Millard-Ball [47]. We reverse the sign for consistency with our other measures. Higher SNDi values mean more disconnected streets, but here we use "negative SNDi" so that a higher value indicates greater connectivity.

Weighted density is the density in each one kilometer square grid cell, weighted by the population of that grid cell (Eq 1). Because the cells are one kilometer square in this study, density is the same as population.

$$weighted\ density = \frac{\sum_i (density_i \times population_i)}{\sum_i (population_i)} = \frac{\sum_i (population_i^2)}{\sum_i (population_i)} for\ each\ grid\ cell\ i \quad (1)$$

To calculate density gradient, we use the negative exponential function, suggested by Mills, in Eq 2, to estimate the intercept and slope parameters.

$$d(r) = d_0 e^{-\alpha r} \quad (2)$$

where d(r) is the population density at distance $r$ from the city center, $d_0$ is the population density at the city center, and $\alpha$ is the density gradient [46, 48, 51, 64]. As the center of each urban center is not known with the available data in this research, and the highest-density pixel is not necessarily the center, we have used an approach that calculates a separate density gradient from every pixel in the urban center. We then assume that the center is the pixel with the largest density gradient. The density gradient is the regression line between density and distance from the center. As with any regression line, it has an intercept (the theoretical density at the

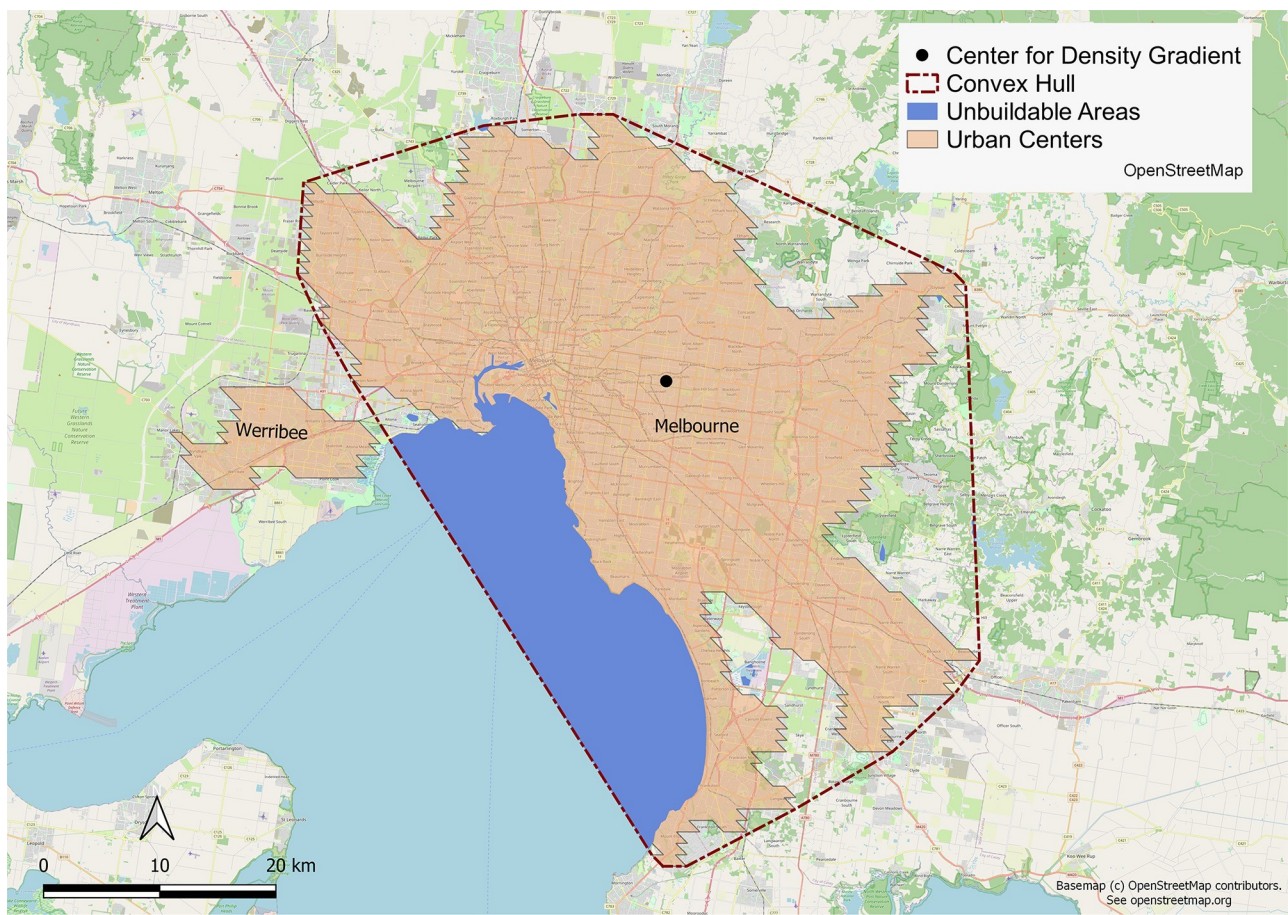

**Fig 2. The convex hull of Melbourne, the unbuildable areas within the convex hull, and the center for density gradient.** Base map and data from OpenStreetMap and OpenStreetMap Foundation.

center), as well as a slope. Therefore, we calculate two different urban form metrics: density gradient slope and density gradient intercept.

The convex hull of all urban centers are calculated, excluding the unbuildable areas around them due to bodies of water as given by the GSHHG shoreline dataset. Compactness is then the ratio of the built area to the convex hull (Eq 3). Fig 2 shows an example.

$$C_{BLT} = \frac{A_{BLT}}{(A_{CVX} - A_{UNB})}$$

(3)

$A_{CVX}$: the area of the convex hull around the developed polygons
$A_{UNB}$: the unbuildable area of the convex hull
$A_{BLT}$: the total area of all built-up polygons
$C_{BLT}$: the built-up coverage (compactness) [12].

## 3.3 Analysis methods

After calculating urban form metrics in PostgreSQL, we assessed the correlations, and the results were plotted as a pairplot with the Seaborn package in Python. As a simple line does not produce a line of good fit in this case, non-parametric or Lowess smoothing was used to model the relationship between the variables that did not fit a predefined distribution and had

a non-linear relationship. After standardizing the data by subtracting the mean and dividing by the standard deviation, we used an unsupervised machine learning algorithm, K-Means clustering, to identify a typology of cities [65]. Cluster analysis is a descriptive tool with no objectively "right" number of clusters $k$. We chose $k$ based on our substantive interpretation, increasing $k$ until clusters were no longer qualitatively distinct. The other machine learning algorithm used in this research is random forest regression, which enables us to separate out the influence of urban form measures and show their associations with outcome variables.

## 4. Results and discussion

We begin with a descriptive analysis of how our different measures of urban form are distributed across various urban centers, how the measures correlate with each other, and how these correlations vary by income group and geographic region.

### 4.1. Correlation between urban form metrics

As might be expected, there is a strong positive correlation between the three density-related measures–weighted density, density gradient intercept, and density gradient slope. This correlation exists across all four income groups. For example, cities with high overall density also tend to have an even denser center and a steep gradient down to the urban periphery. Cities in high-income countries have the lowest weighted density, especially those in Northern America and Oceania, with cities in the other income groups having similar distributions. Cities in Northern America and Oceania also have the lowest density gradient slope and density gradient intercept compared to other areas, but European cities are more aligned with other income groups regarding the density gradient measures. This means that not all high-income countries follow similar trends in their urban form metrics, maybe because of different historical backgrounds and types of planning policies.

The other two measures, compactness and street connectivity, do not correlate as strongly with other metrics, and the correlation is different across income groups. Compactness is correlated with both density gradient metrics in upper-middle and lower-middle-income groups (Fig 3). Cities in Asia, mainly in Eastern and South-Eastern Asia, have the lowest levels of compactness compared to other regions, which means the ratio of built-up areas to the buildable areas in their convex hull is lower. Considering compactness along with density metrics can give us a sense of how sprawled or compact an urban area is.

Street connectivity is correlated with density gradient metrics only in high-income countries, meaning that in these countries, having denser centers is associated with having more connected streets. This measure is lowest in cities located in lower-middle-income countries, especially in South-Eastern Asia and Southern Africa, and highest in high-income countries, particularly in Europe. Compared with cities in the other three income groups, cities in high-income countries display different characteristics. Having a correlation between different metrics can suggest that these metrics can be used interchangeably in those areas and for example if one of them is high, there is a high chance that the other correlated metric is high, too.

### 4.2. Typology of the cities

Cluster analysis captures some aspects of urban form that are not evident on the individual measures and distinguishes the cities with similar characteristics. The K-Means cluster analysis reveals five distinct types of city, as depicted in the radar plot of the cluster centroid (Fig 4). In the first cluster, which we call "High Density", all metrics are high, and except for street connectivity, are the highest of any other group.

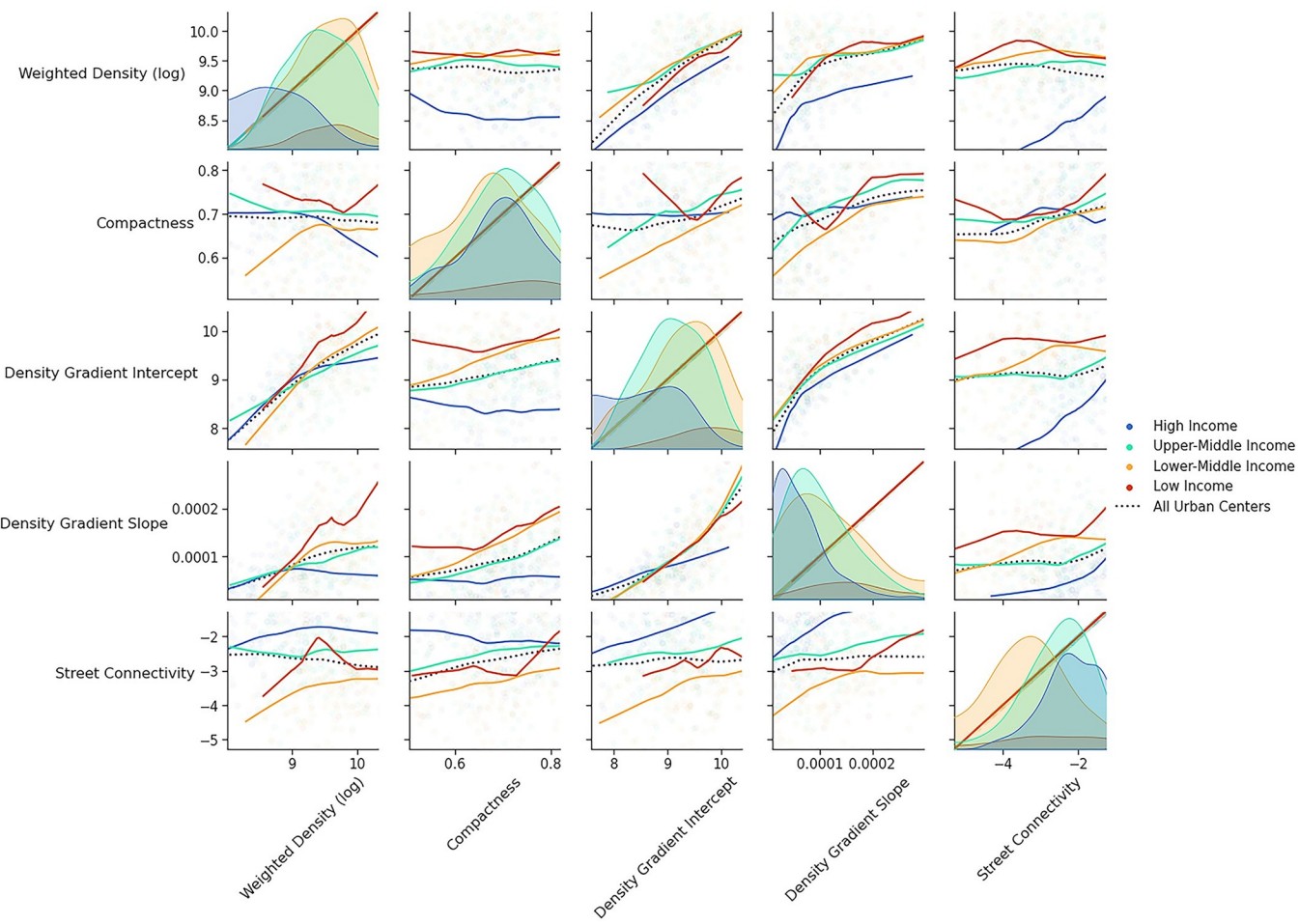

**Fig 3. Correlation of urban form metrics across income groups.**

The second cluster, "Low Density", accounts for 18 percent of the cities and has the lowest weighted density, density gradient slope, and density gradient intercept. Their compactness and street connectivity are moderate, indicating that these low-density cities can also be compact and have connected streets.

The third cluster, "Low Compactness", has the lowest compactness among the clusters. Weighted density of the cities in this group is ranked second highest, and their density gradient and street connectivity is placed in the second lowest rank; showing that high weighted density does not necessarily lead to more compact cities. Agricultural lands, forests, and parks are calculated as potentially buildable areas and in cities like Rio de Janeiro, Brazil, Hong Kong, and Singapore, these protected areas may be one of the reasons for the low compactness metric. In some cities, such as Busan, South Korea, the irregular boundary shape might cause low compactness.

The fourth cluster, "Connected Street Network", includes 36 percent of the cities examined in this study, the largest number among all clusters. It is characterized by highly connected streets. The other metrics are at a moderate to high level compared to other clusters; density gradient and compactness are in the second rank.

The fifth cluster, "Disconnected Street Network"', has very low street connectivity and has the second lowest weighted density and compactness among other groups. The lowest number of cities among all clusters is in this group, with about 11.5 percent of the total large cities.

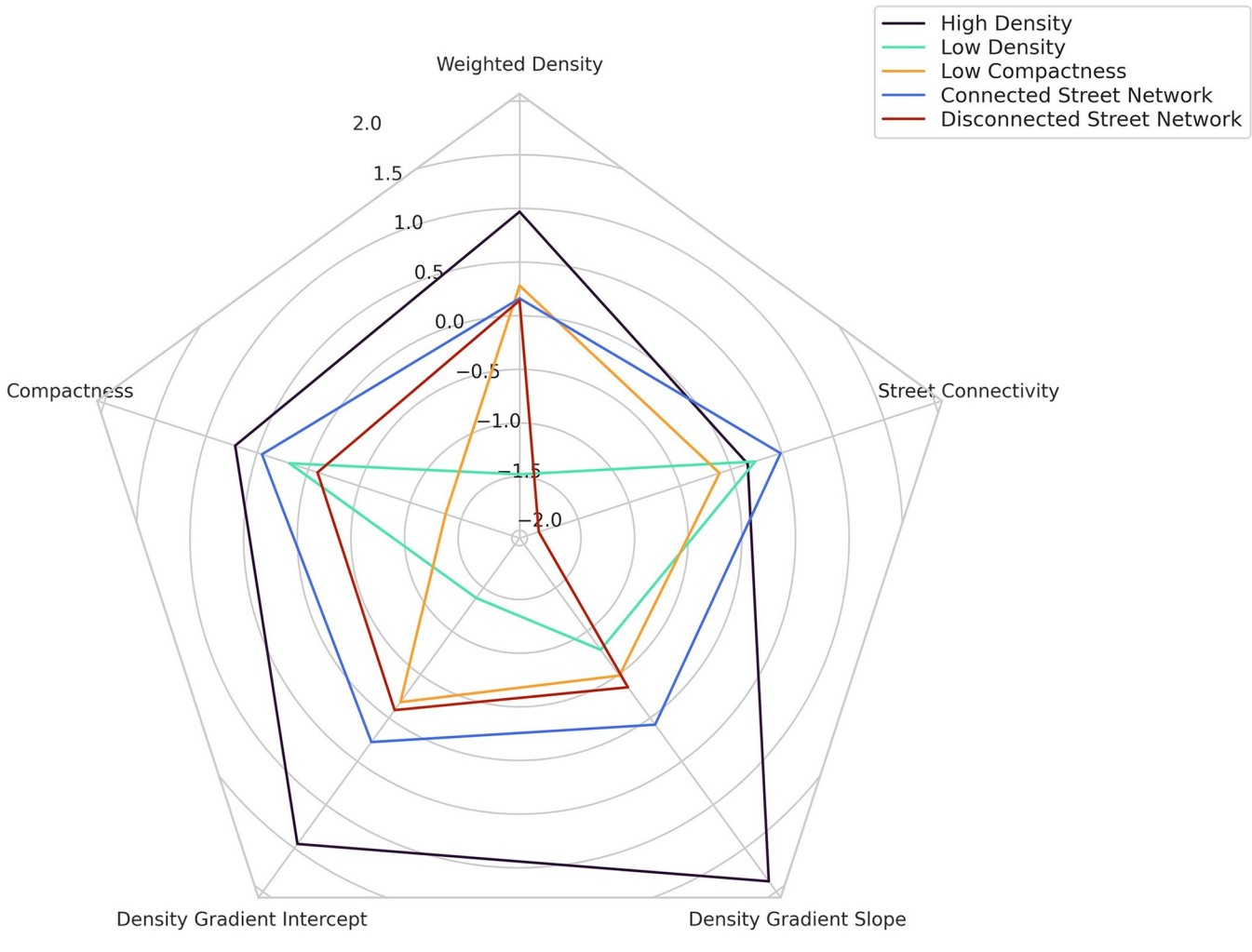

**Fig 4. Radar plot of the cluster centroids.**

## 4.3. Geographic distribution of the clusters

While there is an example of each cluster in almost every world region, there are clear geographic patterns (Figs 5 and 6). The majority of the cities in the "High Density" cluster are located in Asia (71 percent), including 35 percent in India. There is only one European city in this cluster–Brussels, Belgium–which still has the lowest weighted density in this group but one of the highest connected streets among all urban centers in this study. The presence of this European city in the "High Density" group might be related to the postwar large-scale demolition of townhouses for development of the high-rise business district which symbolizes the transformation of Brussels from national capital to Europe's political capital [66]. Brussels' high level of density gradient slope is also related to this transformation which makes it different from most European cities.

75 percent of the cities in the "High Density" cluster are in low-income and lower-middle income countries. Kathmandu, one of the representatives of this cluster (i.e., a city close to the cluster centroid), is located in Nepal, one of the ten fastest urbanizing countries in the world. The traditional core area of Kathmandu consists of a densely built area including narrow streets which gradually shifts towards the periphery surrounded by a vast expanse of

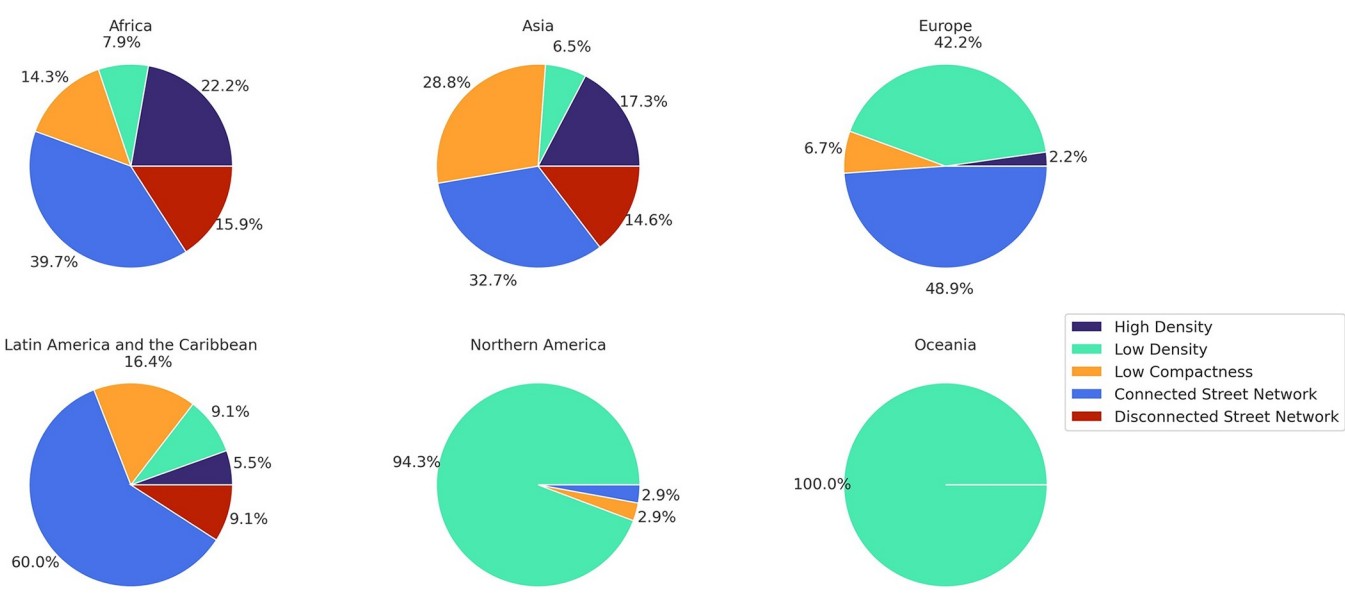

**Fig 5. Distribution of clusters in different geographic regions.**

agriculture-dominated rural area. The population density is increasing rapidly, particularly in the Kathmandu Valley, along the main highways, and close to the border with India [67, 68]. Surat, India, Najaf, Iraq, and Kinshasa, Democratic Republic of the Congo are other representatives of this cluster.

North America, especially the U.S., has the largest portion of cities in the "Low Density" group. Policies such as single-family zoning and automobile-oriented developments are two of the drivers of this low level of density. About a quarter of the cities in this cluster are in Europe, and 21 percent are in Asia (mainly in Eastern Asia and South-Eastern Asia including China, Japan, and Indonesia). 92 percent of the cities in this cluster are in high and upper-middle-

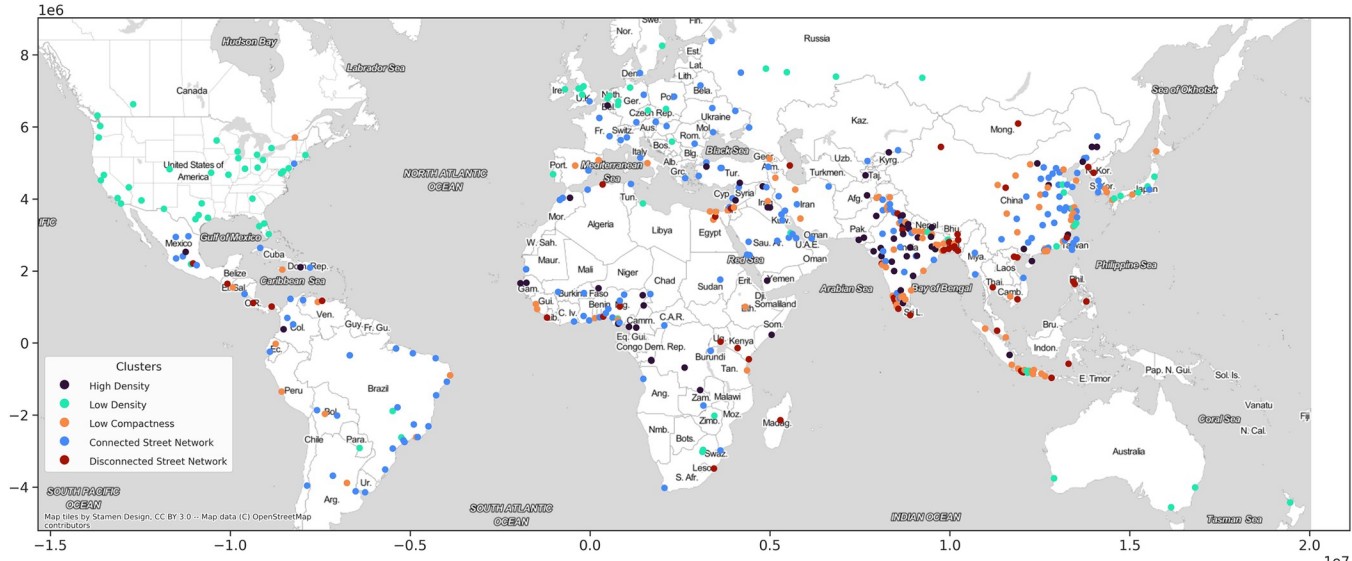

**Fig 6. Typology of cities based on urban form metrics.** Base map and data from OpenStreetMap and OpenStreetMap Foundation.

income countries, suggesting that the income level of the countries affects the density of the cities. Two cities in this group, Indianapolis and Kansas City, both in the U.S., have the lowest levels of weighted density among all the cities studied. Los Angeles, the U.S., Sydney, Australia, Manchester, the U.K., and Toronto, Canada are among the representatives of this cluster.

77 percent of the cities in the "Low Compactness" group are located in Asia (mainly in India, China, and Indonesia). Just one city from Northern America—Montreal, Canada—is present in this group, mainly because of the presence of large green spaces within the convex hull, and the irregular shape of its boundary. Compared to other clusters, "Low Compactness" cities are more evenly distributed across income groups; 56 percent of the cities are in high and upper-middle income and 44 percent in low and lower-middle-income countries. Barcelona and Madrid, Spain, Tehran, Iran, Jakarta, Indonesia, and Lima, Peru are some representatives of this cluster.

More than half of the cities in the "Connected Street Network" cluster are located in Asia, and just one city is in Northern America (New York, U.S.). New York has the lowest density gradient slope and intercept among the cities of this cluster, but has the highest weighted density compared to other Northern American cities overall. While many other U.S. cities such as Boston and Philadelphia have gridiron street patterns in their central areas, they do not also have the relatively high density and compactness that defines cities in the "connected street network" cluster.

Beyond the U.S., about 70 percent of the cities in this cluster are in high- and upper-middle-income countries. Some metropolitan areas such as Istanbul, Turkey, Cape Town, South Africa, São Paulo, Brazil, London, the U.K, Paris, France, Tokyo, Japan, Isfahan, Iran, Rome, Italy, Moscow, Russia, Doha, Qatar, Seoul, South Korea, and Shanghai and Beijing, China are among the representatives of this cluster.

70 percent of the cities in the "Disconnected Street Network" cluster are located in South-Central Asia, South-Eastern Asia, and Eastern Asia, with India, Bangladesh, Indonesia, China, the Philippines, and Vietnam respectively at the highest levels. This cluster has no representatives in Europe, North America, or Oceania. 75 percent of the cities in this cluster are in low- and lower-middle-income countries, indicating that the income level of the countries can influence their street connectivity. One of the main reasons for the presence of South-Eastern Asian cities such as Manila, the Philippines in this cluster are gated communities, which are developer-driven settlements with access centered around the private automobile formed with the rise of a new middle-class in response to fears of crime, inadequate public services, and weak land-use regulation. Poor street connectivity can be seen as a way to generate social and physical exclusivity [15].

In Latin America and the Caribbean cities such as Guatemala City, Panama City, and San Salvador, the combination of weak planning institutions and high rates of crime has resulted in an increase in gated communities. For example, Guatemala City was originally established on a geometric grid under Spanish colonial rule, but has been transformed into a city characterized by informal settlements and by gated communities for the wealthy that are isolated from the rest of the city [15]. The presence of African cities such as Lagos, Nigeria and Kampala, Uganda in this group is mainly due to the fast urbanization rate and large number of informal settlements as well as gated communities in these metropolitan areas [69, 70].

Africa, Latin America and the Caribbean, and China have representatives in all five clusters, but the majority of them are among the "Connected Street Network" group. Most Indian cities are in the "High Density" cluster. Most of the cities in Western Asia are among the cities in the "Connected Street Network" cluster. In South-Central Asia, the connectivity of street networks decreases from west to east. "Disconnected Street Network" and "Low Compactness" clusters represent most of the cities in South-Eastern Asia. No European cities are in the "Disconnected

Street Network" group. The majority of European cities are in the "Connected Street Network" and "Low Density" clusters.

## 4.4. Environmental outcomes of urban form metrics

We now consider how the measures of urban form correlate with two environmental outcomes– access to green space and local air pollution. In addition to the scatter plots, the box plots are used to show the differences between our five clusters and highlight the outliers that are not evident from the scatter plots.

**4.4.1. Urban form and access to green space.** Not surprisingly, green space per capita is negatively correlated with all density metrics (Fig 7). After all, parks, housing with private gardens, and other green spaces will tend to reduce densities. Indianapolis, Kansas City, and Tampa have both the lowest levels of weighted density and the most green space per capita. However, there is no correlation between green space per capita and street connectivity, except in high-income countries. Nor is there a correlation with compactness.

In high-income countries, which have the highest levels of green space per capita worldwide, cities with lower density and less connected street networks tend to have higher green space per capita. In this income group, North American cities have much higher green space per capita compared to European and Asian cities. As shown in Fig 8, green space per capita is higher in the "Low Density" cluster, possibly because single-family zoning in American cities and other sprawling countries provides residents with larger areas of green space in the form of private gardens, but at the expense of reducing densities. The lowest level of green space per capita is found in cities in the "High Density" cluster and also in the countries with lower income, but the other three clusters have similar trends regarding green space per capita.

While in general, there is a negative correlation between density and green space per capita, some outliers suggest that it is possible to have both. Brussels, Belgium is an outlier in the "High Density" group which has a much higher green space per capita compared to the rest of the cities in this group. Based on our results, Brussels is one of the cities with the lowest degrees of sprawl.

A person's access to green spaces is not necessarily determined by the amount of green space per capita, but rather by the percentage of the population that lives in areas with a high green cover. Therefore, we refer to the share of population that resides in high green areas as the "access to green space" metric. This measure is less affected by urban form metrics compared to green space per capita and the difference between various income groups is more significant. Lower-middle-income countries have the highest share of people living in high green areas. Cities in Bangladesh and India such as Noakhali and Thalassery are among the ones with the highest access to green space.

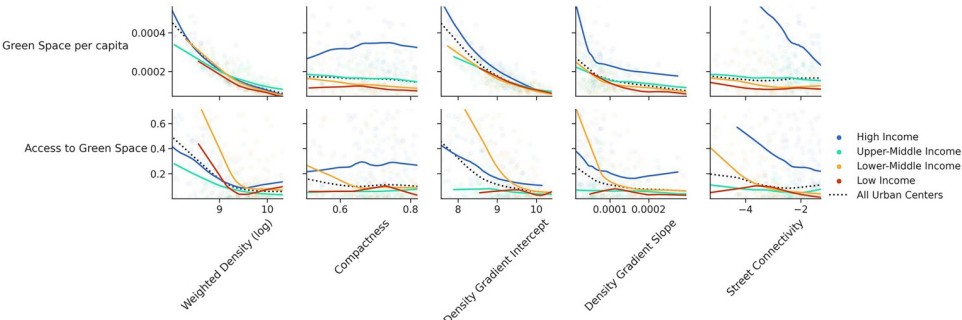

**Fig 7. Correlation of urban form metrics with green space measures across income groups.**

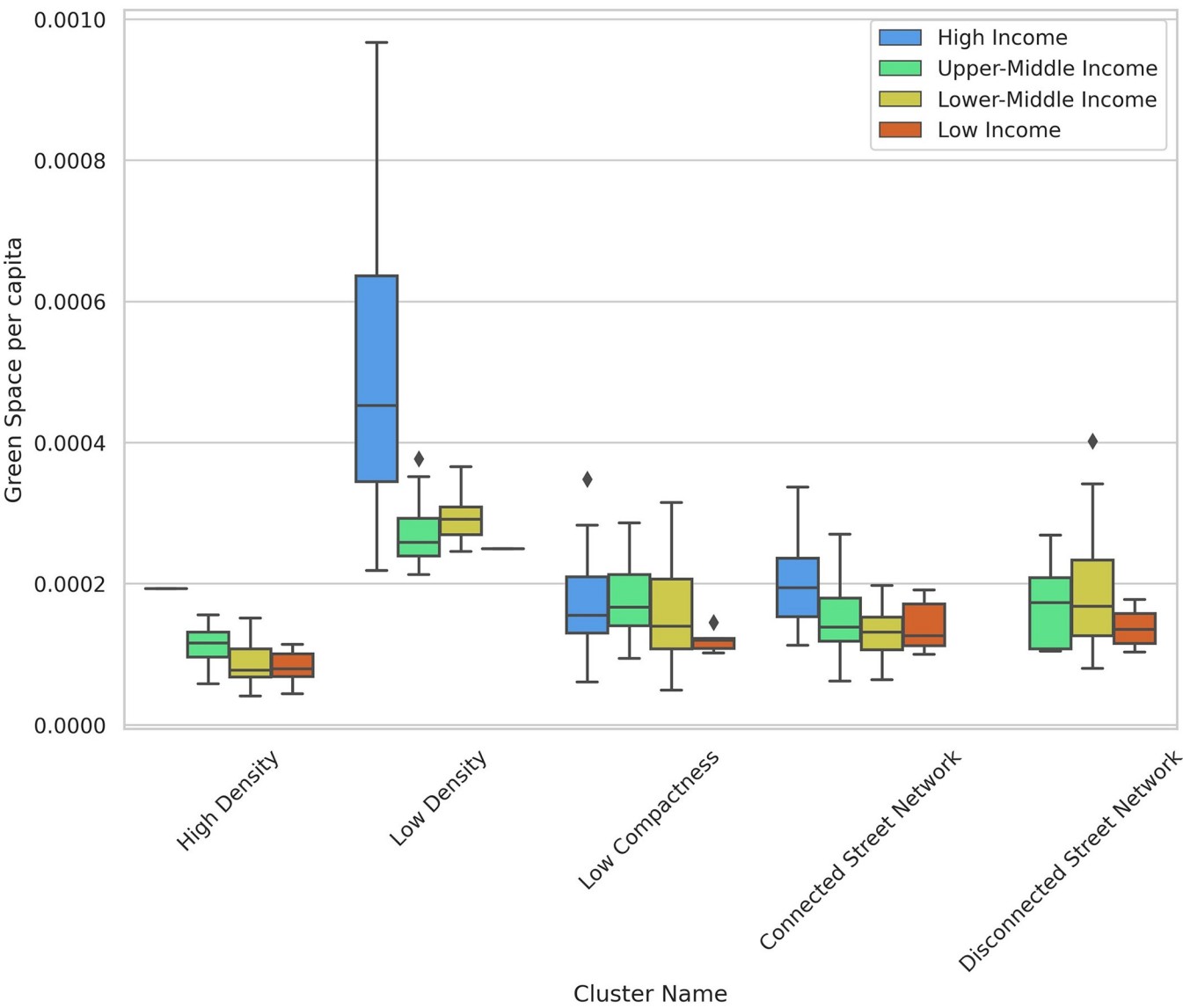

**Fig 8. Box plot—green space per capita vs clusters across income levels.**

"High Density" and "Connected Street Network" clusters have lower levels of access to green space compared to other clusters (Fig 9). However, there are a large number of outliers such as Benin City, Nigeria; Yangon, Myanmar; Havana, Cuba; and Russian and Ukrainian cities from the "Connected Street Network" cluster, as well as Mbuji-Mayi, Democratic Republic of the Congo and Brussels from the "High Density" cluster. These cities have high levels of access to green space, but low green space per capita.

**4.4.2. Urban form and air pollution.** Two air pollution measures are examined in this study: $PM_{2.5}$ transportation emissions per capita and $PM_{2.5}$ concentration (an indicator of exposure). $PM_{2.5}$ transportation emissions per capita is higher in cities of high-income and upper-middle-income countries compared to the rest of the world. According to Fig 10, there is no noticeable relationship between urban form metrics and transportation emissions per capita overall. There is more variation in the emissions exposure between various income

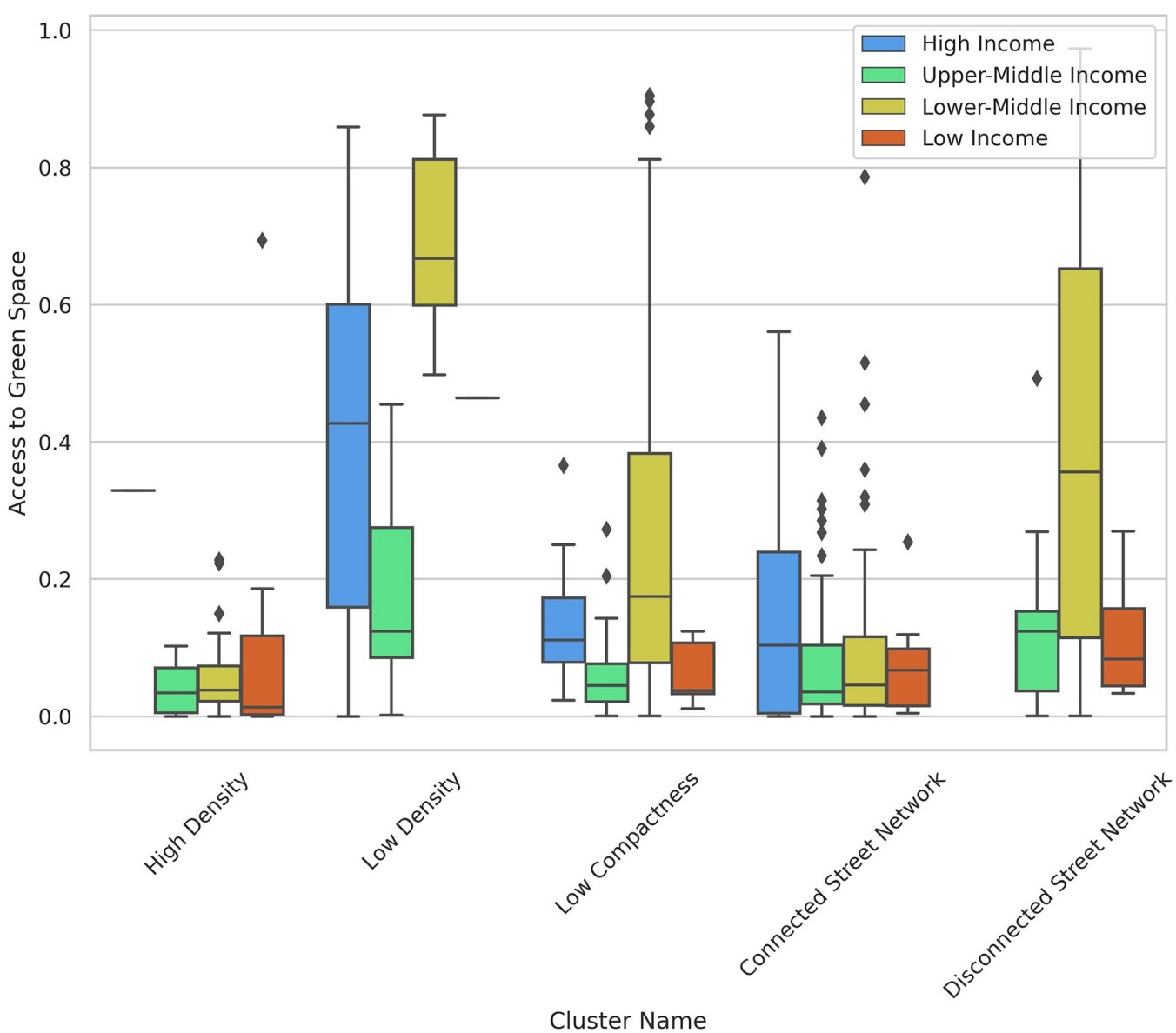

**Fig 9. Box plot—share of population living in high green areas vs clusters across income groups.**

groups. PM$_{2.5}$ concentration is slightly correlated with weighted density in high-income and upper-middle-income groups and has no significant correlations with other urban form measures.

The boxplots illustrate some other aspects of how urban form metrics affect PM$_{2.5}$ measures. Cities with higher density have lower per capita transportation emissions (less driving), but higher PM$_{2.5}$ exposure because more people are living closer to emissions sources. Cities in the "Connected Street Network" cluster have lower transportation emissions per capita, but higher exposure to emissions compared to the cities in the "Disconnected Street Network" (Figs 11 and 12). Taichung, Taiwan is an outlier in the "Connected Street Network" cluster with high per capita transportation emissions. The main outlier in the "Low Density" cluster is Ontisha, Nigeria which has one of the lowest levels of transportation emissions per capita, but highest levels of emissions exposure. Giannadaki et al. [71] reported that most of the emissions

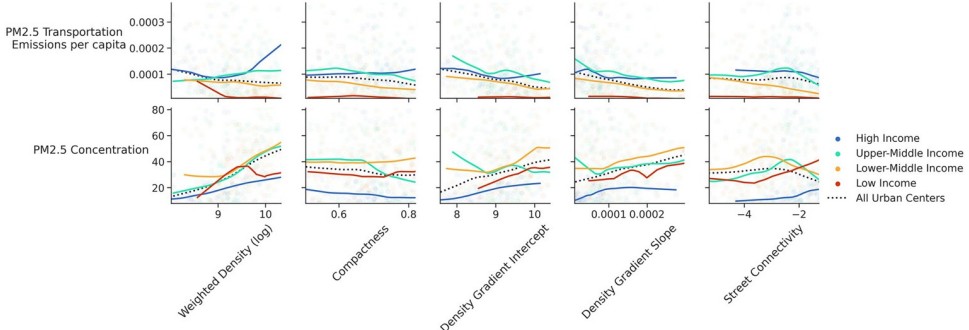

**Fig 10. Correlation of urban form metrics with PM$_{2.5}$ measures across income groups.**

in Nigeria could be linked to natural sources such as Saharan desert dust or wildfires. Wood burning, vehicular emissions, particles generated from cooking using fossil fuels, firewood, industrial emissions, and unsustainable open waste dumpsites are other causes of high levels of emissions exposure in Nigerian cities [72].

The high-income group of the "Low Density" cluster has two outliers with high levels of emissions exposure: Katowice, Poland and Dammam, Saudi Arabia. In Katowice, vehicular emissions account for a considerable part of the total air pollution [73]. Alwadei et al. [74] demonstrate that dust storms account for 42% of PM$_{2.5}$ mass concentration in Dammam. In addition to dust storms, other sources of atmospheric aerosols in Saudi Arabia include heavy oil combustion, resuspended soil, industrial emissions, traffic emissions and marine sources [75]. Cities such as Doha, Qatar, Dubai, United Arab Emirates, and Manama, Bahrain have high levels of PM$_{2.5}$ exposure due to similar drivers.

Among Northern American cities, Vancouver, Calgary, Miami, Portland, and Seattle have the lowest levels of PM$_{2.5}$ concentration. Stockholm, Sweden, Dublin, Ireland, and Copenhagen, Denmark, have the lowest exposure among European cities. These cities follow smart growth policies trying to reduce urban sprawl impacts.

## 4.5. Random forest regression

As a final step, we have used random forest regression that helps with separating out the influence of each measure and showing its association with the outcome variables. When the data has a non-linear trend and extrapolation outside the range of the training data is not important, we can use random forest regression. We used the SHAP value as a united approach to explain the output of our random forest regression model. SHAP assigns each feature an importance value for a particular prediction [76]. On the beeswarm plots (Fig 13), the features are ordered by their predictive power, but we can also see how higher and lower values of the feature will affect the result. Each small dot on the plot represents a single observation. The horizontal axis represents the SHAP value and shows negative or positive effects of the variables, while the color of the point shows us if that observation has a higher or a lower value on the predictor variable, when compared to other observations.

Density (weighted density and density gradient intercept) have the strongest association with green space. High values of weighted density and density gradient intercept have a high negative contribution to the level of green space per capita, while low values of these metrics have a high positive contribution. Similarly, access to green space has a negative correlation with weighted density, density gradient intercept, and street connectivity; however, the impact is less than the impact on green space per capita.

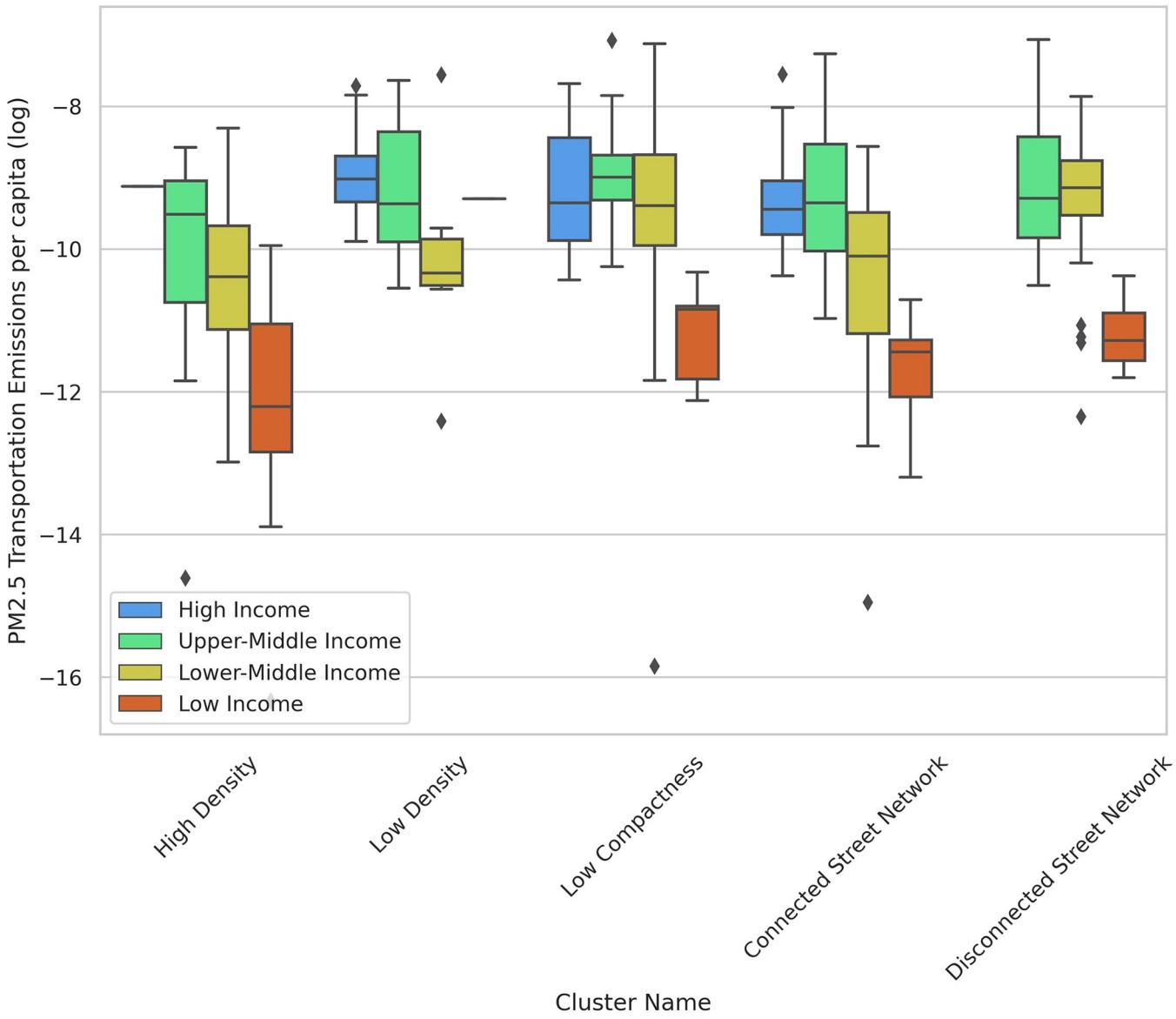

**Fig 11. Box plot—PM$_{2.5}$ transportation emissions per capita vs clusters across income groups.**

Density gradient slope and intercept followed by street connectivity are the best predictors of PM$_{2.5}$ transportation emissions per capita. Higher values of all three reduce PM$_{2.5}$ transportation emissions per capita. More compact cities also lead to lower transportation emissions per capita; however, the impact of compactness is not as high as street connectivity. Moreover, areas with higher weighted density and density gradient slope are likely to be more exposed to PM$_{2.5}$ emissions. The results of the SHAP values are consistent with the earlier analysis.

## 5. Conclusion

Improving our understanding of urban form is essential for progress towards smarter and more sustainable growth. An understanding of urban patterns, dynamic processes, and their relationships is a key objective of urban research, and urban management requires available

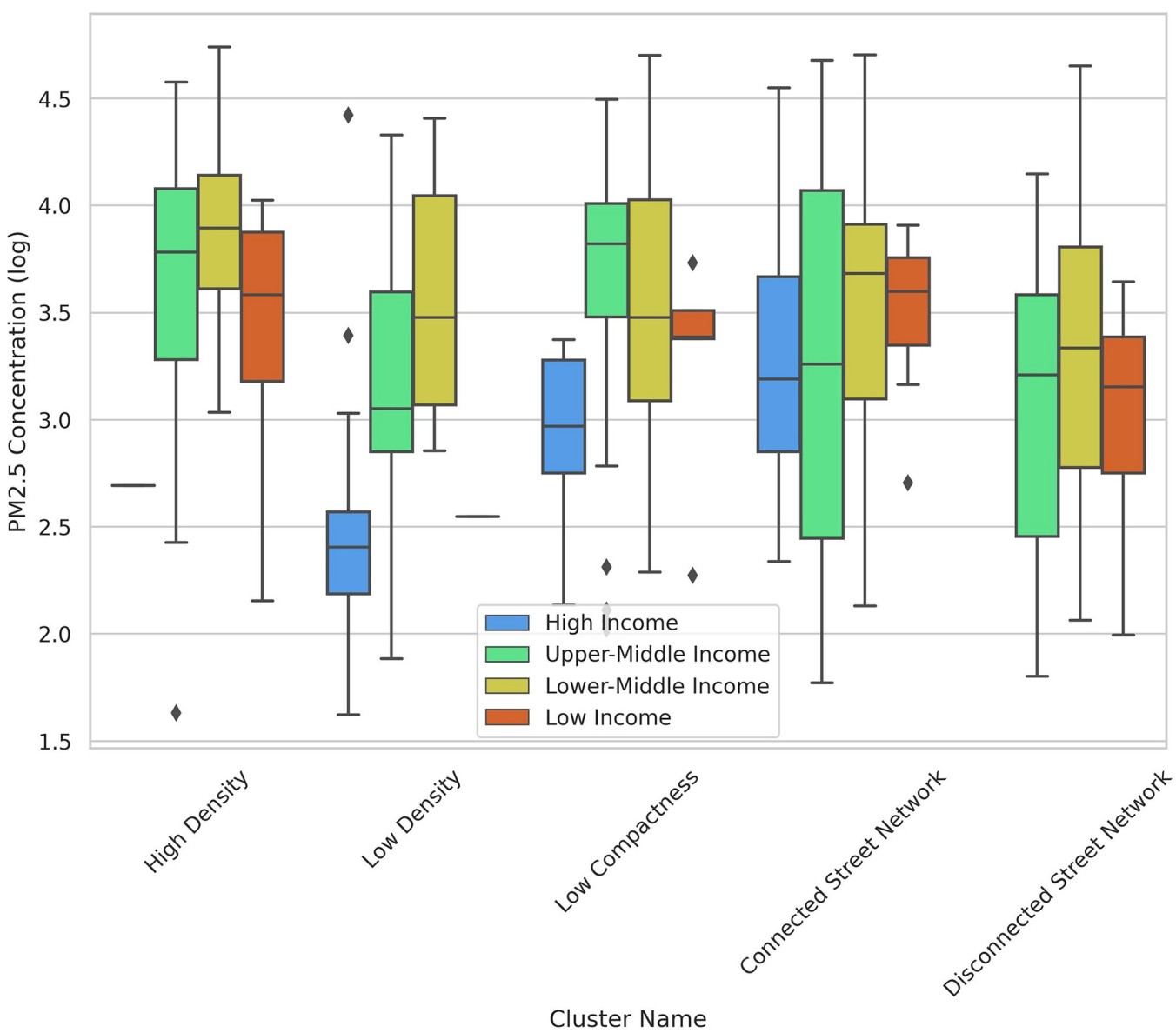

**Fig 12. Box plot—PM$_{2.5}$ concentration vs clusters across income groups.**

and detailed information about the processes and patterns of urbanization. In addition, it allows urban planners and policymakers to decide what policy measures are needed to address problems caused by urban sprawl or to prevent future issues caused by their decisions about urban form.

In this study, we compared various urban form metrics, including weighted density, density gradient (slope and intercept), compactness, and street connectivity, using the GHSL dataset, and examined how they co-vary with each other and across geographic regions and socioeconomic characteristics. The differences in the metrics and their correlations among different geographic regions and income groups show that urban form metrics should be chosen in accordance with the characteristics of cities, and it is inappropriate to generalize results from one income group or geographic region to others.

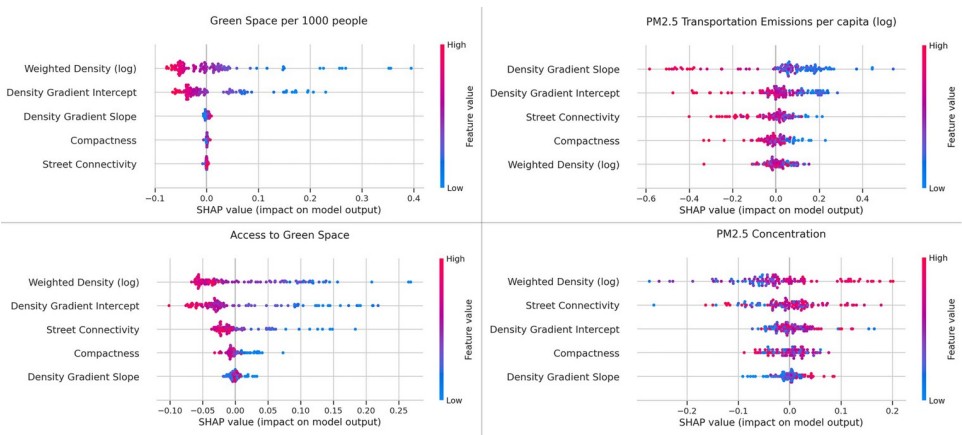

**Fig 13. Random forest regression—SHAP values for all environmental outcome measures.**

We then developed a typology of urban forms, which resulted in five different clusters named "High Density", "Low Density", "Low Compactness", "Connected Street Network", and "Disconnected Street Network". While there is an example of each cluster in almost every world region, there are clear geographic patterns. Perhaps unsurprisingly, "High Density" cities are largely found in Asia and "Low Density" cities in North America, but our results indicate that the majority of "Low Compactness" and "Disconnected Street Network" cities are located in South-Central, South-Eastern and Eastern Asia.

Our study indicates that urban form is correlated with income level and its metrics can vary depending on the country's income level. Further, our findings demonstrate that density does not provide a complete picture of urban form, and multiple indicators are required. Density is often used as a proxy for multiple dimensions of urban form, and our results suggest that this is a reasonable approach in many high-income countries such as the U.S., but not appropriate in other parts of the world where cities can have (for example) high density but low compactness and poorly connected streets.

Density and street connectivity are more helpful for understanding urban form compared to compactness. Although compactness has been widely used as an urban form metric, it is not as informative as density metrics or street connectivity with the prevalent methods used to calculate it. Compactness is also very sensitive to the definition of urban boundary, which makes it more difficult to interpret and generalize. In cases such as Hong Kong, assuming agricultural lands, forests, and parks as potentially buildable areas can lower the compactness. However, in cases like Busan, South Korea, irregular boundary shape might lead to lower levels of compactness.

In the next step, we examined the relationship between our typology and two important policy outcomes: access to green space and air pollution, specifically $PM_{2.5}$. National income and geography have the largest impacts on these two measures, but our focus here is on the impacts of urban form after controlling for these factors. The results show that there is usually a trade-off between density and access to green space. Density (all three variables) have the strongest association with green space. Higher weighted density also increases $PM_{2.5}$ concentration and thus exposure. Street connectivity is associated with lower $PM_{2.5}$ emissions. It is therefore important that future urban planners and policy makers increase green space access in high-density, compact urban areas, which will help alleviate health problems resulting from higher emissions exposure.

Thus, while density is often emphasized as the way to reduce driving and thus $PM_{2.5}$, it comes with a downside: less green space access, and more exposure to $PM_{2.5}$. Moreover, street connectivity actually has a stronger association with reduced $PM_{2.5}$ emissions from the transportation sector, and does not have the same tradeoff with exposure to air pollution or access to green space. A combination of moderate densities and connected streets, therefore, may enable cities to reduce car dependence while maintaining the health and quality of life for their residents.

## Acknowledgments

We are grateful to Christopher Barrington-Leigh for his contributions in providing the street network connectivity dataset and for his excellent comments during the research. We also thank Chris Benner and Carlos Dobkin for their helpful comments on earlier drafts of this paper. This article relies on the Global Human Settlements Layer (GHSL) from the European Commission Joint Research Center.

## Author Contributions

**Conceptualization:** Nazanin Rezaei, Adam Millard-Ball.

**Investigation:** Nazanin Rezaei.

**Methodology:** Nazanin Rezaei, Adam Millard-Ball.

**Software:** Nazanin Rezaei.

**Visualization:** Nazanin Rezaei.

**Writing – original draft:** Nazanin Rezaei.

**Writing – review & editing:** Nazanin Rezaei, Adam Millard-Ball.

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
