## [Decision Letter · Decision Letter 0]

2 Oct 2022

PONE-D-22-22033Urban form and its impacts on air pollution and access to green space:

A global analysis of 462 citiesPLOS ONE

Dear Dr. Rezaei,

Thank you for submitting your manuscript to PLOS ONE. After careful consideration, we feel that it has merit but does not fully meet PLOS ONE’s publication criteria as it currently stands. Therefore, we invite you to submit a revised version of the manuscript that addresses the points raised during the review process.

Please consider all comments of all reviewers including the comments of Reviewer #1

We look forward to receiving your revised manuscript.

Kind regards,

Ahmed Mancy Mosa, Ph.D.

Academic Editor

PLOS ONE

Journal Requirements:

4. We note that Figures 1 and 5 in your submission contain [map/satellite] images which may be copyrighted. All PLOS content is published under the Creative Commons Attribution License (CC BY 4.0), which means that the manuscript, images, and Supporting Information files will be freely available online, and any third party is permitted to access, download, copy, distribute, and use these materials in any way, even commercially, with proper attribution. For these reasons, we cannot publish previously copyrighted maps or satellite images created using proprietary data, such as Google software (Google Maps, Street View, and Earth). For more information, see our copyright guidelines: http://journals.plos.org/plosone/s/licenses-and-copyright.

a) You may seek permission from the original copyright holder of Figures 1 and 5 to publish the content specifically under the CC BY 4.0 license.  

Reviewers' comments:

Reviewer's Responses to Questions

**Comments to the Author**

1. Is the manuscript technically sound, and do the data support the conclusions?

Reviewer #1: No

Reviewer #2: Yes

Reviewer #3: Partly

2. Has the statistical analysis been performed appropriately and rigorously? 

Reviewer #1: No

Reviewer #2: Yes

Reviewer #3: Yes

3. Have the authors made all data underlying the findings in their manuscript fully available?

Reviewer #1: Yes

Reviewer #2: Yes

Reviewer #3: No

4. Is the manuscript presented in an intelligible fashion and written in standard English?

Reviewer #1: No

Reviewer #2: Yes

Reviewer #3: Yes

5. Review Comments to the Author

Reviewer #1: The urban form metrics was calculated and analyzed. The relationship among the urban form metrics, green space and PM2.5 emission were explored too. However, some analysis and conclusion were based on wrong results from wrong equation. So, I recommend rejecting this manuscript for possible publishing in this journal.

The following points should be further improved:

1. There are conflicts between the definition of weighted density and the equation (1). （Weighted density is the density in each one kilometer square grid cell, weighted by the population of that grid cell (Eq 1).）

2. Some indices were used in this article. However, those indices were explained in details. This is not necessary.

Reviewer #2: the manuscript is well written and still needs some edits

1- a flowchart is required to explain the inputs, process and outputs.

2- locations of studied areas are not clear.

3- abstract needs to be concise by explain the main findings

4- introduction doesn't contain a clear reason behind the study and the limitations

Improving our understanding of urban form is essential for progress towards smarter and more sustainable growth (still needs some emphasizes).

The results demonstrate that while higher density is often emphasized as the way to reduce driving and thus PM2.5 emissions (needs extra explanations)

The topic looks interesting, however, it is out of my experience.

all figures with very low resolution therefore, they must be enhanced before accepting such interesting topic.

Reviewer #3: General comments:

Based on 462 cities all over the world, the paper is devoted to exploring the impacts of urban form on air pollution and access to green space. Five indexes (metrics) were employed to characterize urban form, including street connectivity, weighted density, density gradient slope, density gradient intercept, and compactness. A conclusion was drawn as that density is not the only proxy for different aspects of urban form and multiple indicators such as street connectivity are needed.

The paper has its merit, but I am afraid the main conclusions in this paper is not new. In fact, in urban geography, density has never act as the only proxy for different aspects of urban form. In urban studies, a number of shape indexes, spatial entropy, fractal dimension, density decay ratio, and so on, have been employed to characterize urban form.

Specific suggestions

First, density gradient. Density gradient of urban population depends on spatial distribution function. For inverse power law distribution, the density gradient follows scaling law and bear no characteristic length; for negative exponential distribution, the density gradient bear characteristic length. Two types of density gradients have significant different properties. The authors should clarify the related distribution function [See for example, Batty M, Longley PA. Fractal Cities: A Geometry of Form and Function. Academic Press, London, 1994].

Second, compactness. In literature, compactness ratio is defined as the ratio of the built-up area of a city to the area of smallest circle to enclose the city figure. In this paper, the compactness is defined as the ratio of the built-up area of the city to the area of its convex hull. The compactness based on the first definition is more comparable than that based on the second definition. [See for example: Haggett P, Cliff AD, Frey A. Locational Analysis in Human Geography (2nd edition). Edward Arnold, London, 1977]

Third, scaling in compactness of cities. Due to scaling nature of urban form, compactness value depends on the measurement scale or image resolution. The scale dependence of compactness should be discussed in the result analysis. [See for example, Chen Y. Normalizing and classifying shape indexes of cities by ideas from fractals. Chaos, Solitons & Fractals, 2021, 154: 111653]

Fourth, urban shape is one of key element of urban form [see for example, Batty M, Longley PA. Fractal Cities: A Geometry of Form and Function. Academic Press, London, 1994]. A number of shape indexes have been developed by scientists to describe urban form, including compactness ratio, circularity ratio, form ratio, elongation ratio, ellipticity index, and radial shape index [see for example, Chen Y. Normalizing and classifying shape indexes of cities by ideas from fractals. Chaos, Solitons & Fractals, 2021, 154: 111653; Haggett P, Cliff AD, Frey A. Locational Analysis in Human Geography (2nd edition). Edward Arnold, London, 1977]. However, only compactness was made use in this paper. Do you think compactness alone is enough to describe the characteristics of city shape?

Fifth, the validation of multivariate analysis or multi-metric description lies in two conditions: one is orthogonality of variables (indexes, metrics), and the other is dimensional homogeneity of these variables (indexes, metrics). The former ensures the reliability of quantitative explanation, while the latter guarantees the comparability of the calculation results. The opposite of the orthogonality of variables is the collinearity of variables. Since four metrics (indexes) was utilized in this study, at least the Orthogonality of these metrics (indexes) should be clarified or discussed.

Sixth, K-means clustering. Dimensions (unit, order of magnitude, etc) of variables affect the clustering results. K-means clustering is no exception, and its results relies heavily on the dimensions of variables. In order to obtain effective results, it is necessary to standardize or normalize variables. The variable processing should be discussed in the paper.

6. PLOS authors have the option to publish the peer review history of their article (what does this mean?). If published, this will include your full peer review and any attached files.

Reviewer #1: No

Reviewer #2: **Yes: **Mohamed A. E. AbdelRahman

Reviewer #3: No

---

## [Author Response · Author response to Decision Letter 0]

3 Nov 2022

Dear Dr. Mosa,

Thank you for the opportunity to revise the manuscript, “Urban form and its impacts on air pollution and access to green space: A global analysis of 462 cities” for consideration by the PLOS ONE journal. We appreciate the constructive and insightful comments from the reviewers, and believe that the manuscript is considerably improved as a result.

In brief, the major changes in the revision are as follows:

- Added a subsection on the shapes/fractals

- Added a flowchart in the introduction to explain the input, process, and output

- Added the limitations of the study

- Added the references the reviewers have recommended

- Added numerous specific clarifications, reframings, and other changes in line with the specific comments by reviewers

We provide a point-by-point response to each comment in colored text. The manuscript also identifies additions and major revisions in colored text. Note that, for clarity, most minor rewordings and reorderings within a paragraph are not identified this way.

Nazanin Rezaei, Adam Millard-Ball

Comments from Reviewer 1

The urban form metrics was calculated and analyzed. The relationship among the urban form metrics, green space and PM2.5 emission were explored too. However, some analysis and conclusion were based on wrong results from wrong equation. So, I recommend rejecting this manuscript for possible publishing in this journal.

The following points should be further improved:

1. There are conflicts between the definition of weighted density and the equation (1). (Weighted density is the density in each one kilometer square grid cell, weighted by the population of that grid cell (Eq 1).

Response: We have explained in the manuscript “Because the cells are one kilometer square in this study, density is the same as population”. To make the equation more clear, we have separated out density from population and show that they cancel in the updated version of the equation.

2. Some indices were used in this article. However, those indices were explained in details. This is not necessary.

Response: We understand the reviewer is suggesting that less detail is needed in the paper. We agree that there may be too much detail for some readers. However, other readers may not be familiar with the indices. If the editor wishes us to reduce the length of the text, then we would be happy to do so. 

Comments from Reviewer 2

The manuscript is well written and still needs some edits

1- a flowchart is required to explain the inputs, process and outputs.

Response: We have added a flowchart showing the input, process and output in the introduction as below:

2- locations of studied areas are not clear.

Response: All the cities with a population greater than 1 million in the GHSL dataset are studied in this research. Figure 6 maps the cities. When the link to the data is made live, all the names of the cities will be included along with coordinates and other identifying information.

3- abstract needs to be concise by explain the main findings

Response: We would appreciate further clarification from the reviewer about what findings are missing, and what should be made more concise. The abstract is currently 222 words (PLOS ONE guidelines state 250) and the second half summarizes the main findings.

4- introduction doesn't contain a clear reason behind the study and the limitations

Response: There is a policy reason for this study that is explained in the introduction and there is a scientific reason that is our research question. We have included the limitations of the study in the following sections: 

- Sections 2.5, 2.5.1, and 2.5.2: we explained that the lack of access to global land-use and employment datasets prevented us from calculating other urban form metrics such as polycentricity and land-use mix. 

- Section 3.1: we added the limitation that “As the CO2 emission data is available at country level, and not at city level, we did not include it in this study.”

Improving our understanding of urban form is essential for progress towards smarter and more sustainable growth (still needs some emphasizes).

Response: We have added more explanation to the sentence in the introduction as well as two references for more clarification: “progress towards smarter and more sustainable growth can only be made with a complete understanding of urban form and its relationship with air pollution [5,6] and other environmental outcomes, as well as the policy measures required to respond to it.” 

The results demonstrate that while higher density is often emphasized as the way to reduce driving and thus PM2.5 emissions (needs extra explanations)

Response: We have added more explanation in the introduction as well as four clarifying references: “Finally, we conclude by suggesting that while higher density is often emphasized as the way to reduce driving [36,37] (due to higher access to active and public transit) [38,39] and thus PM2.5 transportation emissions, it comes with a downside: less green space access, and more PM2.5 exposure.” 

The topic looks interesting, however, it is out of my experience.

Response: Thank you.

all figures with very low resolution therefore, they must be enhanced before accepting such interesting topic.

Response: We have submitted the figures with high resolution based on the PLOS ONE requirements. The resolution might be low in the versions that were embedded in the text due to the way that the PDF was compiled. 

Comments from Reviewer 3

General comments:

Based on 462 cities all over the world, the paper is devoted to exploring the impacts of urban form on air pollution and access to green space. Five indexes (metrics) were employed to characterize urban form, including street connectivity, weighted density, density gradient slope, density gradient intercept, and compactness. A conclusion was drawn as that density is not the only proxy for different aspects of urban form and multiple indicators such as street connectivity are needed.

The paper has its merit, but I am afraid the main conclusions in this paper is not new. In fact, in urban geography, density has never act as the only proxy for different aspects of urban form. In urban studies, a number of shape indexes, spatial entropy, fractal dimension, density decay ratio, and so on, have been employed to characterize urban form. 

Response: We agree with the reviewer that there is a lot of literature on why density is not enough to demonstrate urban form. However, most of these analyses have been small-scale. We study the cities of the entire world. Our findings complement the results of the previous studies and provide more insights in emphasizing the importance of focusing on other metrics such as street connectivity. If there are other studies that have similar work that we have not mentioned, we would be happy to reference them. We respond in more detail about the other measures of urban form (e.g., shape indexes) below.

Specific suggestions

First, density gradient. Density gradient of urban population depends on spatial distribution function. For inverse power law distribution, the density gradient follows scaling law and bear no characteristic length; for negative exponential distribution, the density gradient bear characteristic length. Two types of density gradients have significant different properties. The authors should clarify the related distribution function [See for example, Batty M, Longley PA. Fractal Cities: A Geometry of Form and Function. Academic Press, London, 1994].

Response: We acknowledge there are other methods to calculate the density gradient. We used the negative exponential, as it is more straightforward to calculate and has been validated in numerous empirical studies. We referenced the source you provided to clarify that our approach is not the only choice. 

Second, compactness. In literature, compactness ratio is defined as the ratio of the built-up area of a city to the area of smallest circle to enclose the city figure. In this paper, the compactness is defined as the ratio of the built-up area of the city to the area of its convex hull. The compactness based on the first definition is more comparable than that based on the second definition. [See for example: Haggett P, Cliff AD, Frey A. Locational Analysis in Human Geography (2nd edition). Edward Arnold, London, 1977]

Response: This is another issue where there are different choices, and we have clarified the alternatives in the text. Many cities are not circular, for example Montreal, because of topography and water constraints. The method we have used is more flexible to account for different shapes. There is a trade-off between using the circular method and the method we have used. 

Third, scaling in compactness of cities. Due to scaling nature of urban form, compactness value depends on the measurement scale or image resolution. The scale dependence of compactness should be discussed in the result analysis. [See for example, Chen Y. Normalizing and classifying shape indexes of cities by ideas from fractals. Chaos, Solitons & Fractals, 2021, 154: 111653]

Response: We appreciate the reviewer raising this point. As they note, compactness depends on the image resolution, and we now note in the paper that this will affect the absolute value of our compactness measure. However, we are mostly interested in the difference in compactness between the cities in the sample, and since we use a consistent image resolution, we expect scale dependence to have limited impact on our results. 

Fourth, urban shape is one of key element of urban form [see for example, Batty M, Longley PA. Fractal Cities: A Geometry of Form and Function. Academic Press, London, 1994]. A number of shape indexes have been developed by scientists to describe urban form, including compactness ratio, circularity ratio, form ratio, elongation ratio, ellipticity index, and radial shape index [see for example, Chen Y. Normalizing and classifying shape indexes of cities by ideas from fractals. Chaos, Solitons & Fractals, 2021, 154: 111653; Haggett P, Cliff AD, Frey A. Locational Analysis in Human Geography (2nd edition). Edward Arnold, London, 1977]. However, only compactness was made use in this paper. Do you think compactness alone is enough to describe the characteristics of city shape?

Response: Compactness is definitely not enough to explain all aspects of urban form. 

For future work, we could use a wider range of urban form measures, such as the ones the reviewer has mentioned. We have also added material on the measures you recommended in Section 2 of our paper. However, our focus is on measures of urban form that have a theoretical or empirical link with the environmental outcomes that we study, rather than the broader literature on urban shapes that the reviewer references. We believe that the measures listed above are likely to have minimal impacts above and beyond what is already captured by compactness (although this certainly could be a topic for future study). 

Fifth, the validation of multivariate analysis or multi-metric description lies in two conditions: one is orthogonality of variables (indexes, metrics), and the other is dimensional homogeneity of these variables (indexes, metrics). The former ensures the reliability of quantitative explanation, while the latter guarantees the comparability of the calculation results. The opposite of the orthogonality of variables is the collinearity of variables. Since four metrics (indexes) was utilized in this study, at least the Orthogonality of these metrics (indexes) should be clarified or discussed.

Response: The correlation matrix of the variables used in this study is shown in Figure 3. In this study, we have used machine learning methods which do not rely on orthogonality of the variables and do not require input features to be orthogonal. 

Sixth, K-means clustering. Dimensions (unit, order of magnitude, etc) of variables affect the clustering results. K-means clustering is no exception, and its results relies heavily on the dimensions of variables. In order to obtain effective results, it is necessary to standardize or normalize variables. The variable processing should be discussed in the paper.

Response: Thank you for the detailed and constructive suggestions. In the revised version of the manuscript (page 21), we clarify that we standardized the variables before applying the algorithm. We are unsure if the reviewer had other concerns regarding the variable processing, but we hope we have addressed the issue. 

Journal Requirements

Response: We will make the code available on acceptance and before publication. 

Response: We will make the data available on acceptance and before publication. 

4. We note that Figures 1 and 5 in your submission contain [map/satellite] images which may be copyrighted. All PLOS content is published under the Creative Commons Attribution License (CC BY 4.0), which means that the manuscript, images, and Supporting Information files will be freely available online, and any third party is permitted to access, download, copy, distribute, and use these materials in any way, even commercially, with proper attribution. For these reasons, we cannot publish previously copyrighted maps or satellite images created using proprietary data, such as Google software (Google Maps, Street View, and Earth). For more information, see our copyright guidelines: http://journals.plos.org/plosone/s/licenses-and-copyright.

a) You may seek permission from the original copyright holder of Figures 1 and 5 to publish the content specifically under the CC BY 4.0 license. 

Response: Thank you for these resources. The source of the background map of both figures 1 and 5 is OpenStreetMap which is open source and free to use. The license agreement is available here: https://www.openstreetmap.org/copyright and we have added the source in the captions of the figures based on the instructions on this link. We have also added the source of the basemap within figures 1 and 5 (which are referred to as figures 2 and 6 in the revised version).

To our understanding, OSM does not provide any letters of permission, but refers users to this copyright agreement. We assume that PLOS One staff have experience with OpenStreetMap-based figures and would appreciate any further guidance on how to explain the copyright.

---

## [Decision Letter · Decision Letter 1]

14 Nov 2022

Urban form and its impacts on air pollution and access to green space:

A global analysis of 462 cities

PONE-D-22-22033R1

Dear Dr. Rezaei,

We’re pleased to inform you that your manuscript has been judged scientifically suitable for publication and will be formally accepted for publication once it meets all outstanding technical requirements.

Kind regards,

Ahmed Mancy Mosa, Ph.D.

Academic Editor

PLOS ONE

Additional Editor Comments (optional):

Reviewers' comments:

Reviewer's Responses to Questions

**Comments to the Author**

1. If the authors have adequately addressed your comments raised in a previous round of review and you feel that this manuscript is now acceptable for publication, you may indicate that here to bypass the “Comments to the Author” section, enter your conflict of interest statement in the “Confidential to Editor” section, and submit your "Accept" recommendation.

Reviewer #1: All comments have been addressed

Reviewer #3: All comments have been addressed

2. Is the manuscript technically sound, and do the data support the conclusions?

Reviewer #1: Yes

Reviewer #3: Partly

3. Has the statistical analysis been performed appropriately and rigorously? 

Reviewer #1: No

Reviewer #3: Yes

4. Have the authors made all data underlying the findings in their manuscript fully available?

Reviewer #1: Yes

Reviewer #3: Yes

5. Is the manuscript presented in an intelligible fashion and written in standard English?

Reviewer #1: Yes

Reviewer #3: Yes

6. Review Comments to the Author

Reviewer #1: All the comments had been responds and revised. No other shortcomings were found this time. So I recomend to accept this article this time.

Reviewer #3: The quality of this paper has been improved after modification. The authors dealt with my review comments to a great degree. To my thinking, the paper is acceptable for publication in PLoS ONE.

7. PLOS authors have the option to publish the peer review history of their article (what does this mean?). If published, this will include your full peer review and any attached files.

Reviewer #1: No

Reviewer #3: No

---

## [Editor Report · Acceptance letter]

28 Dec 2022

PONE-D-22-22033R1 

Urban form and its impacts on air pollution and access to green space:
A global analysis of 462 cities 

Dear Dr. Rezaei:

I'm pleased to inform you that your manuscript has been deemed suitable for publication in PLOS ONE. Congratulations! Your manuscript is now with our production department. 

Kind regards, 

on behalf of

Dr. Ahmed Mancy Mosa 

Academic Editor

PLOS ONE